# Aqueous Extract of Sea Squirt (*Halocynthia roretzi*) with Potent Activity against Human Cancer Cells Acts Synergistically with Doxorubicin

**DOI:** 10.3390/md20050284

**Published:** 2022-04-23

**Authors:** Yuting Zhu, Shanhao Han, Jianhui Li, Hongwei Gao, Bo Dong

**Affiliations:** 1Sars-Fang Centre, MoE Key Laboratory of Marine Genetics and Breeding, College of Marine Life Sciences, Ocean University of China, Qingdao 266003, China; yutingzhu19@hotmail.com (Y.Z.); hanshanhao@stu.ouc.edu.cn (S.H.); lijianhui@stu.ouc.edu.cn (J.L.); 2Laboratory for Marine Biology and Biotechnology, Qingdao National Laboratory for Marine Science and Technology, Qingdao 266237, China; 3Technology Center of Qingdao Customs, Qingdao 266002, China; 4Institute of Evolution & Marine Biodiversity, Ocean University of China, Qingdao 266003, China

**Keywords:** ascidian, *Halocynthia roretzi*, anti-tumor, drug combination, doxorubicin

## Abstract

Marine ascidian is becoming one of the main sources of an antitumor drug that has shown high bioactivity and extensive application in cancer treatment. *Halocynthia roretzi*, an edible marine sea squirt, has been demonstrated to have various kinds of biological activities, such as anti-diabetic, anti-hypertension, and enhancing immunity. In this study, we reported that aqueous extracts from the edible parts of *H. roretzi* presented significantly inhibiting the efficiency on HepG-2 cell viability. The separate mixed compound exhibited strong effects of inhibitory proliferation and induced apoptosis via the generation of ROS along with the concurrent loss of mitochondrial membrane potential on tumor cells. Furthermore, we found that there existed a significantly synergistic effect of the ascidian-extracted compound mixture with the anti-cancer drug doxorubicin. In the presence of the extracts from *H. roretzi*, the dose of doxorubicin at the cellular level could be reduced by a half dose. The extracts were further divided by semipreparative-HPLC and the active ingredients were identified as a mixture of fatty amide, which was composed of hexadecanamide, stearamide, and erucamide by UHPLC-MS/MS. Our results suggest that the potential toxicity of ascidian *H. roretzi* in tumor cells, and the compounds extracted from *H. roretzi* could be potentially utilized on functional nutraceuticals or as an adjunct in combination with chemotherapy.

## 1. Introduction

Cancer, the world’s second biggest killer disease, is the uncontrolled growth of abnormal cells which starts almost anywhere in the human body [1]. Chemotherapeutics entering the blood circulation system non-selectively target actively proliferating cells, ultimately leading to the damage of both healthy and tumor cells [2]. To avoid this problem, one possible approach is to develop novel and more efficient antitumor agents. Over the decades, natural products have shown many advantages in cancer treatment, such as novel chemical structures, low toxicity and side effects [3,4]. Furthermore, natural products from marine organisms with a wide range of sources, extraordinary chemical structures and brilliant biological activities have attracted greater amounts of scientific attention. So far, more than 36,000 compounds have been isolated from marine sources. The most abundant sources of active marine products are fungi, sponges, and tunicates [5,6].

Ascidians (tunicates, sea squirts) are sessile invertebrates belonging to the phylum chordata, which represent a source of bioactive natural products [7]. With the deeper insight into tunicates, over 1000 active compounds have been isolated, and some of them have been used as clinical drugs and preclinical leads [8]. ET-743 (Trabectedin), a milestone in the development of marine drugs, was initially isolated from the ascidian *Ecteinascidia turbinata*. ET-743 binds the minor groove of DNA, triggers DNA cleavage, and affects the tumor microenvironment. Because of these mechanisms, ET-743 can inhibit the neoangiogenesis and the metastatic of tumor cells, which are important in cancer therapy [9]. Furthermore, this is the first anti-tumor drug derived from ocean animals and was approved in 2015 by the FDA for the treatment the soft tissue sarcoma in the United States [10]. Aplidine (dehydrodidemnin B) is a translation elongation factor (eEF1A2) inhibitor that was isolated from tunicate *Aplidium albicans* [11]. The results of clinical trials proved that it showed lower myelotoxicity than didemnin B, the first antineoplastic agent isolated from ascidians entering clinical trials [12] and high activity against medullary thyroid carcinoma, renal-cell carcinoma, and melanoma [13]. Therefore, investigation on ascidians is important for the discovery of new drugs with novel chemical scaffolds.

*Halocynthia roretzi (H. roretzi)* is an edible marine sea squirt, which is mainly composed of three parts: the tunic, the inner soft and orange-colored body tissues (the edible part), and the stolon (Figure 1A). *H. roretzi* is widely cultured in Japan and South Korea, and used as commercial seafood, because of its rich nutritional content such as eicosapentaenoic acid (EPA), docosahexaenoic acid (DHA), carotenoids, taurine, plasmalogen and other micronutrients [14,15]. In recent years, there have been many reports about the active components from *H. roretzi*, including findings that peptides and hemagglutinin from *H. roretzi* exhibited antioxidant and antimicrobial activity [16,17]. Hence, it is important to investigate the pharmacological properties of *H. roretzi*.

In this study, we reported that the aqueous extract of *H. roretzi* tissue showed a strong inhibitory effect on the growth and proliferation of tumor cells. Moreover, after a series of separations of the extract, the separated product had a certain broad spectrum inhibitory activity on several tumor cells and a synergistic effect with doxorubicin. Consequently, we have characterized and fractionated an *H. roretzi* extract and identified the active ingredients of the extract. Our study provides a new direction for the discovery of active substances in ascidians and provides a theoretical basis for *H. roretzi* as a nutrient combined chemotherapy treatment of cancer.

## 2. Results

### 2.1. The Small Molecular of Water Extract from Ascidian Tissue Inhibited the Proliferation of HepG-2 Cells

We dissected adult of *H. roretzi* and separated its body into three parts, including body tissues (edible parts), tunic, and stolon (Figure 1A). Water was used to extract the bioactive ingredients from tissues of *H. roretzi.* Three kinds of water extracts from body tissues (WB-H), from the tunic (WT-H), and from the stolon (WS-H) tissues of *H. roretzi* were analyzed for their antitumor effect by MTT assay and using human hepatocellular carcinoma HepG-2 cells. The results showed that WB-H exhibited more remarkable antiproliferative effects than that of the other two extracts (Figure 1B and Appendix A). We further found that the antiproliferative effects of WB-H on HepG-2 cells was dose- and time- dependent (Figure 1C,D). To figure out whether the active ingredient is concentrated in the macromolecule or small molecule fraction, dialysis was used for the preliminary separation of WB-H extract. As shown in Figure 2A, the larger molecules were trapped inside the dialysis tubing, while the smaller molecules passed through the tubing and entered the water solution outside. The viability of HepG-2 cells treated with these two samples were detected by MTT assay. After 48 h of incubation, a small molecule fraction, at a concentration of 1 mg/mL and 2 mg/mL, reduced the cell viability to 40% and 18%, respectively (Figure 2B). However, a macromolecule sample did not show a high anti-proliferation effect on HepG-2 cells (Appendix A). The mouse fibroblast cell line L929, a normal cell line, was used to detect the toxicity of the water extracts. Moreover, the toxicity of small molecules to mouse fibroblast L929 cells is far less than that to HepG-2 cells (Figure 2B). These results suggest that the small molecule part of water extract from *H. roretzi* tissue is significantly effective on the inhibition of HepG-2 cell proliferation, meanwhile it is less toxic to the normal L929 cells.

### 2.2. A Group of Small Molecular Components of WB-H Inhibited the Tumor Cell Proliferation

The small molecule part of WB-H was further separated by silica gel column chromatography. The separation conditions were optimized by using various ternary mixed solvents as eluent. As shown in Figure 3A, a silica gel column chromatography eluted by the ternary system of ethyl acetate, methanol and formic acid, and gradient elution modes was performed. The sample WB-H was then separated into six fractions (WB-H-S1 to WB-H-S6). These six groups were further evaluated for their anticancer activities against HepG-2 cells by MTT assay. The results showed that the first group (WB-H-S1) showed a significant inhibitory activity against HepG-2 cells (Figure 3B and Appendix A). To investigate the effect of WB-H-S1 in vitro, several tumor cell lines including human hepatocellular carcinoma HepG-2, human melanoma cell BF16F1, human fibrosarcoma cell HT-1080, human thyroid anaplastic carcinoma cell BHT-101, and human cervical carcinoma cell HeLa were employed to detect the antitumor activities. As shown in Figure 3C, the proliferation of the tested tumor cells was affected by WB-H-S1 in a dose-dependent manner, especially in concentrations of 300 μg/mL and 400 μg/mL. Conversely, WB-H-S1 showed significantly less toxicity to L929 cells at higher concentrations (Figure 3C). The HepG-2 cell growth was also inhibited after 72 h treatment with WB-H-S1 (Figure 3D). These data demonstrated that WB-H-S1 exhibited broad-spectrum and long-term anticancer activity.

### 2.3. WB-H-S1 Induced the Apoptosis of HepG-2 Cells

Apoptosis is an important mechanism of cell death in cancer drug screening which is divided into extrinsic and intrinsic pathways. The annexin V-FITC (AV) and propidium iodide (PI) double-staining method is capable of providing information of necrotic cells, including the early- and late-stage of cell apoptosis. The potential of WB-H-S1 to induce apoptosis was analyzed by flow-cytometry after staining HepG-2 cells with AV and PI. Figure 4A,B show that in the control, the percentage of alive cells was 82.31 ± 2.90%, whereas the treatment with 200 µg/mL WB-H-S1 for 24 h led to a decrease in the percentage of alive cells to 60.91 ± 8.44%. Meanwhile, the percentage of early- and late- apoptotic cells increased from 17.22 ± 2.95% to 38.41 ± 8.86% (Figure 4A,B).

Bcl-2 is a member of the anti-apoptotic class of B cell leukemia-2 gene product family proteins that regulate the intrinsic mitochondrial pathway of cell apoptosis [18]. Consistent with results of AV-PI staining, compared to the control, increasing the concentration of WB-H-S1 up to 200 μg/mL for 24 h decreased the expression of anti-apoptosis factor Bcl-2 protein, while it increased the expression of apoptotic factor Bax protein (Figure 4C,D). These results clearly demonstrated that WB-H-S1 induced apoptosis in HepG-2 cells. In addition, the expression of the p53 protein was significantly increased (Figure 4C,D). These results suggested that WB-H-S1 induced HepG-2 cell apoptosis by activating the expression of the p53 protein. As a further support to this conclusion, we found that the presence of pifithrin-α (PFT-α), which was used as an inhibitor of p53, reduced the mortality induced by WB-H-S1 (Figure 4E).

### 2.4. WB-H-S1 Induced of ROS Generation and Disturbed Mitochondrial Membrane Potential

Intracellular reactive oxygen species (ROS) were monitored after the treatment with WB-H-S1 by 2,7-dichlorodihydrofluorescein diacetate (DCFH-DA) dye, as ROS play an important role in the regulation of various cellular processes. Higher concentrations of ROS is toxic for tumor cells and induces cell apoptosis [19]. Figure 5A,B showed that ROS generation was significantly enhanced after treatment with 200 µg/mL WB-H-S1 for 24 h. Previous studies showed that ROS generation was closely related to potential changes in the mitochondrial membrane (MMP) [20]. The intrinsic mitochondrial pathway of apoptosis is regulated by MMP. To further investigate the mitochondrial-mediated pathway in apoptosis induced by WB-H-S1, a fluorescence microplate was used to measure MMP in WB-H-S1 treated cells through JC-1 staining. As shown in Figure 5C, when subjected to 200 µg/mL WB-H-S1 for 24 h, this resulted in the decline of MMP.

### 2.5. WB-H-S1 Acted Synergistically with the Doxorubicin on the Inhibition of the Growth of Cancer Cells

The simultaneous therapy with conventional chemotherapeutics and natural compounds increasingly shows its advantages in cancer treatment [4]. We assayed the synergistic effects of doxorubicin with WB-H-S1. Doxorubicin is a topoisomerase II (TOP2) inhibitor that works by intercalating DNA and further inhibiting DNA replication [21]. It is a highly effective chemotherapy drug which has been used in clinics for years. However, the use of doxorubicin has been limited as a result of the risks presented by its cardiotoxicity [22]. WB-H-S1 at a range of concentrations (50, 70, 90, 100, and 150 μg/mL) was combined with doxorubicin at two specified concentrations (5 μg/mL and 10 μg/mL) for 48 h, cell viability was detected by the MTT assay. As shown in Figure 6A,B, all range of concentrations of the combination of WB-H-S1 with doxorubicin enhanced the anti-proliferative activity of doxorubicin, and doxorubicin combined with WB-H-S at concentration of 10 μg/mL (Figure 6B) exerted better synergistic effects than that at a concentration of 5 μg/mL (Figure 6A). The results showed that in the presence of WB-H-S1, the anti-proliferative activity of doxorubicin was significantly enhanced. When HepG-2 cells and L929 cells were treated with WB-H-S1 at concentrations of 100 μg/mL, for 48 h in the presence or absence of doxorubicin, a significantly higher cytotoxicity towards HepG-2 cells than L929 cells was observed (Figure 6C). These results suggest that, at the cellular level, WH-B-S1 not only acts synergistically with the doxorubicin, but also reduces the dose of doxorubicin, and thus it may decrease the damage of the chemotherapeutics to the normal cells. Collectively, these data support the potential therapeutic application of WB-H-S1 in combination with doxorubicin.

### 2.6. The Active Ingredients of WB-H-S1 Were Identified as Mixture of Several Fatty Amides

In order to find the active ingredient in WB-H-S1, we carried out a systematic fractionation of extracts to purify and identify the ingredients that are responsible for its antiproliferative activity. Preparative-HPLC led to the isolation of twelve ingredients from WB-H-S1 (Figure 7A) and the fractions corresponding to the peaks, a-l (Figure 7A), were collected, and the solvent was removed. The samples were then redissolved in methanol. In addition to the main peak, the rest of the eluting portion was also collected. All components isolated by semipreparative-HPLC were assayed for their antitumor activity by MTT experiments. The results showed that compounds from peak e and peak j presented inhibitory activity against HepG-2 cells (Figure 7B and Appendix A). A synergism assay was performed by combining the compounds from peak e or peak j with doxorubicin, respectively at the concentration of their IC50. The results of the synergism assay showed that peak j showed stronger synergies than peak e with the doxorubicin on the inhibitory activity against HepG-2 cells (Appendix A). Therefore, the compounds of peak j were identified by UHPLC-MS/MS further through the comparison of retention times and the fragmentation mass spectra with the standard substances (Appendix A). The results showed that compounds in peak j contained erucamide, stearamide, and hexadecanamide (Figure 7C). Among them, erucamide showed significant inhibition on cell proliferation for HepG-2 cells (Appendix A). These results indicate that fatty acid amides from *H. roretzi* show potent anti-tumor activity and act synergistically with doxorubicin on tumor cell proliferation.

## 3. Discussion

The consumption of seafood is beneficial for human health, as seafood is a valuable dietary component that includes n-3 polyunsaturated fatty acids, anserine, taurine, iodine, selenium, vitamin A, vitamin K, vitamin D, tocopherols, B vitamins, astaxanthin, etc. [23]. In this study, we attempted to monitor the antitumor ability of *H. roretzi*, which is widely cultured and consumed as a foodstuff. Water, as a non-toxic extractant, is often used to extract the ingredients in medicinal organisms. Our result clearly showed that *H. roretzi* extract inhibited the proliferation of HepG-2 cells in a dose- and time-manner. We then performed a preliminary separation by dialysis and phase silica gel. After enrichment, the cell viability assay showed that the active component WB-H-S1 inhibited the growth of several cancer cell lines. WB-H-S1 presented a lower cytotoxic effect against normal cells and a more selective one towards tumor cells. The differential effect of WB-H-S1 in different cell lines may result from specific genetic expression profiles, membrane proteins, adhesion molecules in the surface, and/or other characteristics which make them more sensitive to WB-H-S1.

Apoptosis is programmed cell death and occurs in various physiological events. Therefore, apoptosis is the most common direction towards anticancer drugs to exert their anticancer effect. Fucoidan extracted from *Sargassum cinereum* exerts a potent anticancer effect against caco-2 cells by inducing cell apoptosis [24]. In the present study, HepG-2 cells treated with 200 µg/mL WB-H-S1 for 24 h were stained with AV/PI, and cell apoptosis was determined by flow cytometry. The population of early apoptotic cells increased from 17.2 ± 2.95% to 38.4 ± 8.86%, confirming that WB-H-S1 induced apoptosis in the HepG-2 cell apoptotic death activation by intrinsic pathway is closely related to mitochondrial membrane depolarization [25]. The balance between the expression levels of Bcl-2 and Bax, two major members of the Bcl-2 family proteins, plays a critical role in triggering apoptosis via the mitochondria intrinsic pathway [26]. WB-H-S1 also increased the expression of Bax protein while reducing the expression of Bcl-2 in HepG-2 cells. In addition, we also found that WB-H-S1 induced a loss of mitochondrial membrane potential in HepG-2 cells. These results further confirmed that the WB-H-S1 activated tumor cell apoptosis through the mitochondrial pathway.

Cancer cells are characterized by ROS overproduction as result of hypermetabolism. High ROS levels in cancer cells promotes cancer development by participating in signaling pathways [27]. Therefore, tumor cells are more sensitive to changes in ROS levels than normal cells. Recently, induced oxidative stress causing severe damage in cancer cells by increasing ROS and/or inhibiting antioxidant processes was proposed as a parameter to be considered in antitumor drug screening [28]. Selenocompounds exert anticancer effects by inducing ROS generation resulting ROS-induced cellular damage [29]. Moreover, anthracycline-based drugs, such as doxorubicin, can induce the accumulation of oxidative stress and ultimately lead to cell death [30]. In the present study, increased ROS generation after WB-H-S1 treatment suggested that ROS accumulation contributed to cell death induced by WB-H-S1. Recent studies have revealed that ROS acts as a signal triggering p53 activation, and the levels of cellular ROS are modulated by p53 in many ways [31]. Therefore, we also measured the level of p53 protein in the cells after the treatment of WB-H-S1. As expected, WB-H-S1 increased intracellular p53 protein levels, and cell mortality was reduced in the presence of a p53 inhibitor. Increased ROS and p53 protein levels contribute to the antitumor activity of the WB-H-S1, and this is likely the main reason of the observed low toxicity in normal cells.

For the characterization of the active ingredients of the *H. roretzi* extract, the UHPLC-MS/MS method was developed. Peak j was identified as a mixture of hexadecanamide, stearamide, and erucamide by comparing with standards. It is worthy of note that we found that a mixture of fatty amides from *H. roretzi* shows an inhibitory effect on tumor cell proliferation and have a synergistic effect with doxorubicin. Fatty acid amides as the endogenous lipid bioregulators are important signaling molecules participating in locomotion, angiogenesis, and many other processes [32]. Their high biological activity and receptor-mediated mechanisms of action, suggest that these compounds may be a new class of biological regulatory molecules [33]. The antitumor activity testing of three fatty acid amides revealed that erucamide, and not hexadecanamide or stearamide, was toxic to HepG-2 cells. Erucamide is a potent acetylcholinesterase inhibitor and is a nontoxic natural product found in vegetables, particularly radish [34]. Erucamide has been reported to be useful against memory deficits induced by Alzheimer’s and has the potential to antagonize depression and anxiety [33]. But the antitumor activity of erucamide is less well known. Here we proved that *H. roretzi* extracts containing erucamide could induce HepG-2 cell apoptosis. It has been reported that polyunsaturated fatty acids (PUFA) such as DHA and EPA have the potential effects against diverse types of cancers. DHA has been demonstrated to have an anti-proliferative effect on human MDA-MB-231 breast cancer cells [35]. EPA exhibited a cytotoxic effect on the colorectal cancer cell lines Caco-2, HT-29, HCT116, LoVo, SW480, and SW620 in vitro [36]. Polyunsaturated fatty acid amides (PUFA amides), the active product of fatty acid hydrolysis, show similar anticancer activities. Some PUFA amides like N-docosahexaenoylethanolamine and N-eicosapentaenoylethanolamine exert immune-modulating effects and inhibit breast cancer growth in vitro and in vivo [37]. Fatty acid amides which derived from oleic acid, palmitic acid, EPA, and DHA can induce cell apoptosis in the SKOV-3 ovarian cancer cell line [38].

To overcome the problems related to the adverse effects induced by chemotherapeutic drugs, cancer patients are given a combination of drugs which produce improved outcomes with fewer side effects [39]. PUFA and PUFA amides in combination with antitumor drugs have the potential to increase the sensitivity of tumor cells to conventional therapies. DHA intensively enhanced the cytotoxicity of cisplatin and significantly reduced its effective concentrations both in vitro and in vivo [40]. DHA also has been reported to improve the efficacy and reduce the therapy-associated side effects of doxorubicin [41]. Fatty acid amide derivatives of doxorubicin were more lipophilic and exhibited more anti-proliferative activity in ovarian and colon cell lines [42]. In this study, *H. roretzi* aqueous extracts also showed synergistic effects combined with doxorubicin. The generation of ROS is an important molecular mechanism through which doxorubicin induces cell death [43]. The effects of WB-H-S1 and doxorubicin on ROS accumulation may be complementary. These may be one of the reasons why *H. roretzi* aqueous extracts enhanced the efficacy of doxorubicin. Based on all the experimental results, the combination of the extracts and doxorubicin may reduce the toxicity of doxorubicin in clinical treatment. The application of drug combinations with doxorubicin and WB-H-S1 needs to be supported by more experimental data in vivo. Further work to study the molecular mechanism of drug combinations and the evaluation of its antitumor activity by xenograft models with tumor cells needs to be undertaken. Our findings thus supported the endeavor to analyze and fractionate the *H. roretzi* extracts with the intent of identifying potential anticancer ingredients.

## 4. Materials and Methods

### 4.1. Materials

Sea squirts (*H. roretzi*) were purchased from the XUSHAN group (Weihai, Shandong, China). Dulbecco’s Modified Eagle Medium (DMEM), fetal bovine serum (FBS) and phosphate-buffered saline (PBS) were purchased from Biological Industries (Kibbutz Beit Haemek, Israel). MTT (3-(4, 5-dimethylthiazol-2-yl)-2,5-diphenil tetrazolium bromide) was purchased from Sigma-Aldrich (St. Louis, MO, USA). Doxorubicin and Gemcitabine were purchased from Aladdin Biotech (Shanghai, China). All commercial chemicals were used without further purification.

### 4.2. Preparation of Sea Squirt (H. roretzi) Extract

Three tissues of *H. roretzi* were shattered into small pieces. The tissue was decocted for 60 min with three volumes of distilled water. After the decoction, the supernatant was recovered and concentrated by rotary evaporation. The sea squirt water extracts were stored at −70 °C until use.

### 4.3. Cell Culture

HeLa cells and BHT-101 were cultured in DMEM containing 10% FBS and 1% penicillin–streptomycin. HepG-2 cells and HT-1080 cells were cultured in MEM containing 10% FBS and 1% penicillin–streptomycin. B16F1 cells were cultured in RPMI1640 containing 10% FBS and 1% penicillin–streptomycin. All cell lines were cultured in 37 °C in a 5% CO_2_ incubator cell growth medium that was changed each day and cells were subcultured with 0.25% trypsin-ethylenediaminetetraacetic acid solution when they were 80–90% confluent.

### 4.4. MTT Assay for Cell Viability

HepG-2 cells were cultured in 96-well plates, 1 × 10^4^ cells per well. Cells were incubated with extracts for 48 h. After incubation for 48 h, medium was removed and 100 µL of MTT work solution (0.5 mg/mL) in DMEM was added and kept for 4 h at 37 °C in a CO_2_ incubator. The formazan crystals were dissolved in DMSO and optical density (OD) value was measured at 490 nm using a microplate reader (GENios, TECAN, Männedorf, Switzerland).

### 4.5. Dialysis

A Biotech Cellulose Ester (CE) Membrane (MWCO: 100 Da, Spectrumlabs, Los Angeles, CA, USA) was used for dialysis. WB-H (20 mL) were loaded into dialysis tubing. Dialysis tubing containing the samples was immersed in distilled water (1 L) overnight. The external solution of dialysis tubing was the solution of small molecules which was collected and evaporated to dryness under 95 °C and reduced pressure, then dissolved with MeOH. The internal solution of dialysis tubing consisted of solutions of macromolecular compounds.

### 4.6. Preparation of Silica Gel Fractionation of Water Extracts of H. roretzi

A small fraction (3 g) of water extract was collected after dialysis and subjected to normal-phase silica gel (EtOAc-MeOH-CH2O3 (5:1:0.02→4:1:0.02 →3:1:0.02→2:1:0.02→1:1:0.02→MeOH), and six fractions (WB-H-S1 to WB-H-S6) were obtained. WB-H-S1 to WB-H-S6 were evaporated to dryness under 55 °C and reduced pressure, then dissolved with MeOH to get six test solutions. Solutions were stored at 4 °C.

### 4.7. Apoptosis

HepG-2 cells were cultured in 6-well plates (1 × 10^6^ cells per well) and treated with WB-H-S1 for 24 h. The apoptotic analysis was performed with an Annexin V-FITC apoptosis detection kit (Beyotime Institute of Biotechnology, Shanghai, China) following the manufacturer’s instructions. Briefly, WB-H-S1-treated cells were harvested and washed twice with PBS. The cell suspensions were incubated with Annexin V-FITC (1/40, *v*/*v*) and PI (1/20, *v*/*v*) for 15 min at room temperature. After labeling, cells were washed once with PBS and stained cells were analyzed using a flow cytometer (FC 500, Beckman Coulter, Brea, CA, USA). A total of 30,000 cells were collected in flow cytometry to detect red and green fluorescence, respectively.

### 4.8. Western Blot

HepG-2 cells were pretreated with 200 µg/mL of the WB-H-S1 for 24 h incubation. Total protein was obtained using a lysis buffer (P0013, Beyotime Institute of Biotechnology) with protease and phosphatase inhibitors (P1045, Beyotime Institute of Biotechnology). Next, HepG-2 cells were centrifuged to remove precipitation. Polyacrylamide gels (12%) were used to load the proteins sample, protein separated depend on molecular weight by using SDS-PAGE and the gels were electro-transferred onto polyvinylidene difluoride membranes (Millipore, Burlington, MA, USA). The membranes were blocked by 5% skim milk in TBST and then incubated with the primary antibodies at 4 ℃ overnight. Subsequently, the membranes were washed with TBST three times and treated with the appropriate secondary antibodies for 1 h. Image J software (NIH, Bethesda, MD, USA) was used to analyze the band intensities.

### 4.9. Determination of Cellular ROS Generation

Intracellular ROS levels were quantified by measuring the fluorescence intensity of the permeable probe 2′,7′-dichlorofluorescein diacetate (DCFH-DA), which was converted into fluorescent dichlorofluorescein (DCF) by intracellular ROS. HepG-2 cells cultured in 24-well plates (1 × 10^5^ cells per well) were treated with WB-H-S1 in 200 µg/mL for 24 h at 37 °C for 24 h, and then the cells were washed three times with a DCFH-DA staining buffer. Cells were incubated with DCFH-DA (10 µM, Beyotime Institute of Biotechnology) at 37 °C for 30 min. Cellular fluorescence intensity was measured by fluorescence microplate (Tecan Infinite 200 Pro, TECAN, Zurich, Swiss). Alive cells were quantified by MTT staining. The average fluorescence intensity in each well was the ratio of the value result of the microplate and MTT staining. Decreased values compared to the control were considered to represent decreases in intracellular ROS levels.

### 4.10. Mitochondrial Membrane Potential

Variation of mitochondrial membrane potential (ΔΨm) was measured using JC-1 (Beyotime Institute of Biotechnology) according to the manufacturer’s instructions. HepG-2 cells were cultured in 24-well plates (1 × 10^5^ cells per well) and treated with WB-H-S1 at concentration of 200 µg/mL for 24 h, then cells were incubated with the JC-1 probe solution (5 µg/mL) at 37 ℃ incubator for 30 min. Cells were washed twice with JC-1 staining buffer. The fluorescence intensities of JC-1 monomers (green) and aggregates (red) were detected by fluorescence microplate (Tecan Infinite 200 Pro). The ∆ψ m for each treatment group was calculated as the ratio of aggregates (red) to monomers (green) fluorescence.

### 4.11. Isolation and Identification of Active Ingredients

Fingerprint characterization and fractionation of extracts was done by semipreparative HPLC. The chromatographic separation was performed on a Hitachi L-2000 Semi-preparative-HPLC (Hitachi, Tokyo, Japan) with a Kromasil 100-5-C18 column (10 × 250 mm) (Akzo Nobel, Bohus, Sweden). The extract was dissolved in methanol and filtered through a 0.45 µm membrane filter before injection. The flow rate was 2 mL/min and the injection volume was 200 µL. The photodiode-array detector was set at 210 nm for absorbance tracing. The elution gradient consisted of eluent A (deionized, ultrapure water) and eluent B (methanol). Initial conditions were 95% eluent A and 5% eluent B, followed by a linear gradient to 10% eluent B for 10 min, then a linear gradient to 20% eluent B for 25 min. This was followed by a linear gradient to 70% eluent B for 30 min, until a final linear gradient to 100% eluent B for 50 min. The fractions corresponding to the major peaks, a-l, were collected and the solvent was removed by evaporation. Structures were identified by UHPLC-MS/MS by comparison of retention times, mass spectra, and fragmentation mass spectra with reference standards. The UHPLC-MS/MS analysis was performed on an UltiMate 3000 (Thermo Fisher Scientific, Cleveland, OH, USA) coupled with a Q Exactive (Thermo Fisher Scientific). The flow rate was 300 µL/min and the injection volume was 10 µL. The mobile phase consisted of eluent A (ultrapure water + 0.01% formic acid+2 mM ammonium acetate) and eluent B (acetonitrile + 0.01% formic acid+2 mM ammonium acetate). The gradient (in *v*/*v*%) started with 10% of B for 1 min. Then B increased linearly to 100% for 5 min. This composition was maintained for an additional 3 min and then returned to 10% of B for 10 min. The column was kept at a temperature of 40 °C. The MS/MS analysis was performed in the positive atmospheric pressure ESI mode and multiple-reaction monitoring (MRM) detection mode. The precursor ion at *m*/*z* 256.3 and the product ion at *m*/*z* 88.0 were selected for the identification of hexadecanamide from extracts. The precursor ion at *m*/*z* 284.3 and the product ion at *m*/*z* 88.0 were selected for the identification of stearamide from extracts. The precursor ion at *m*/*z* 338.4 and the product ion at *m*/*z* 83.1 were selected for the identification of erucamide from extracts.

### 4.12. Statistical Analysis

All data were presented as mean ± SEM. A Student’s t-test was used to assess statistical significance between groups. The *p*-value < 0.01 and *p*-values < 0.05 were considered as significant different and different, respectively.

## Figures and Tables

**Figure 1 marinedrugs-20-00284-f001:**
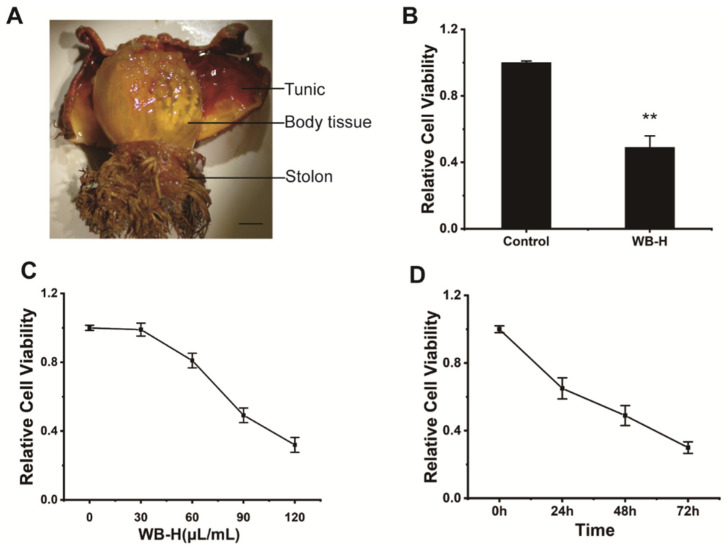
WB-H inhibited the proliferation of HepG-2 cells. (**A**) *H. roretzi* was dissected into three parts (tunic, body tissue, stolon). (**B**) HepG-2 cells were treated with water extracts from body tissue of *H. roretzi* or distilled water (Control) at a concentration of 100 μL/mL for 24 h. Relative cell viability measured by MTT assay. (**C**) Relative viability of HepG-2 cells following WB-H treatment at a range of concentration (30, 60, 90, 120 μL/mL) for 48 h. (**D**) Relative viability of HepG-2 cells following WB-H treatment at the concentrations of 100 μL/mL for 24, 48 and 72 h, respectively. Data are mean ± SEM, *n* = 3, ** *p* < 0.01.

**Figure 2 marinedrugs-20-00284-f002:**
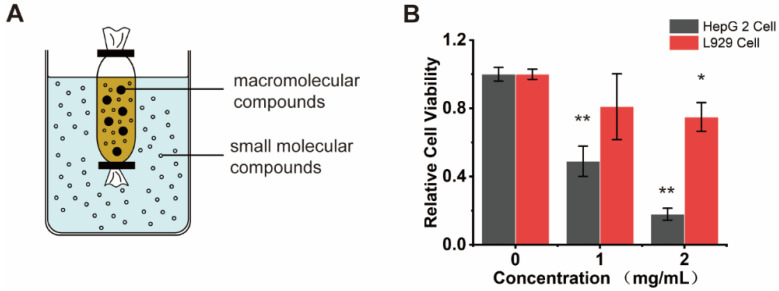
Small molecule component of WB-H inhibited the proliferation of HepG-2 cells. (**A**) Molecular porous membrane tubing with molecular weight cut-offs and diameters of 100 D was used to separate large and small molecules of WB-H. (**B**) HepG-2 cells and L929 cells were treated with small molecule part of WB-H at concentration of 1 mg/mL and 2 mg/mL for 48 h. Relative cell viability was measured by MTT assay. Data are mean ± SEM, *n* = 3, * *p* < 0.05, ** *p* < 0.01.

**Figure 3 marinedrugs-20-00284-f003:**
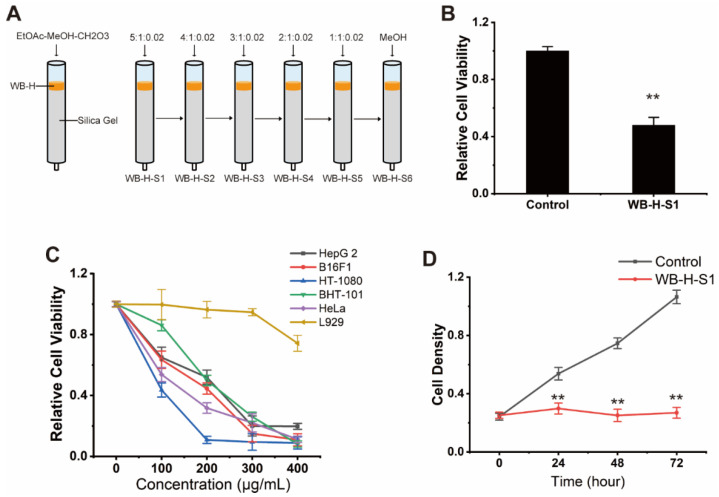
Antiproliferative effect of WB-H-S1 against human cancer cells. (**A**) A small molecular part of WB-H was divided into six components by silica column chromatography, which was eluted by the ternary system of ethyl acetate, methanol and formic acid, isocratic and gradient elution modes to be performed. (**B**) HepG-2 cells were treated with WB-H-S1 at a concentration of 200 μg/mL or treated with methanol (control). Relative cell viability was measured by MTT assay. (**C**) Relative viability of five human tumor cells that were treated with a range of concentrations (100, 200, 300, and 400 μg/mL) for 48 h. (**D**) Relative viability of HepG-2 cells following WB-H-S1 treatment at the specified concentrations of 200 μg/mL or treated with methanol (Control) for 24, 48, and 72 h, respectively. Data are mean ± SEM, *n* = 3, ** *p* < 0.01.

**Figure 4 marinedrugs-20-00284-f004:**
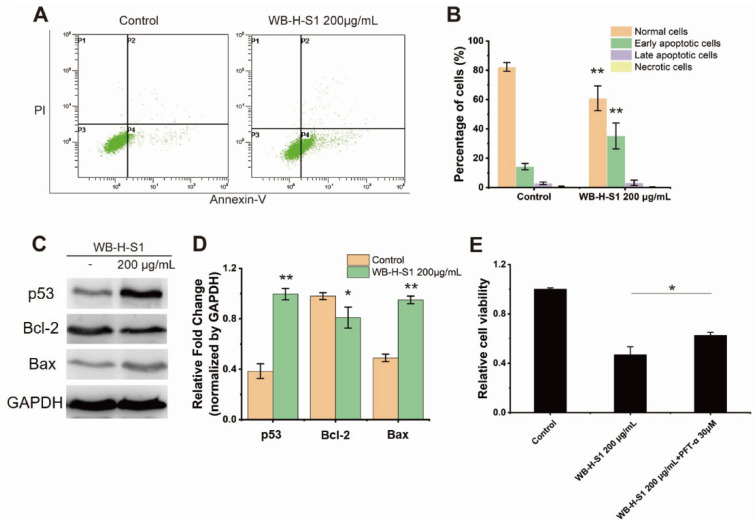
WB-H-S1 induced HepG-2 cell apoptosis (**A**) HepG-2 cells were treated with WB-H-S1 at a concentration of 200 µg/mL or with methanol (Control) for 24 h. AV-/PI-stained cells were determined by flow cytometry (P1, necrosis cells; P2, late apoptotic cells; P3, normal cells; P4, early apoptotic cells). (**B**) The percentage of normal cells, early apoptotic cells, late apoptotic cells, and necrosis cells after 24 h treatment with WB-H-S1 at concentration of 200 µg/mL or with methanol (control). (**C**) HepG-2 cells were treated with WB-H-S1 at a concentration of 200 µg/mL or with methanol (Control) for 24 h. Protein levels of p53, Bcl-2, and Bax were examined by western blot. (**D**) Statistics of the relative expression levels of p53, Bcl-2, and Bax proteins. (**E**) HepG-2 cells- were exposed to WB-H-S1 at the concentrations of 200μg/mL for 48 h in the presence or absence of PFT-α. Relative cell viability was measured by MTT assay. Data are mean ± SEM, *n* = 3, * *p* < 0.05, ** *p* < 0.01.

**Figure 5 marinedrugs-20-00284-f005:**
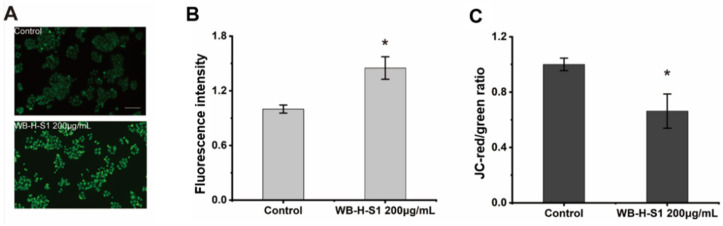
WB-H-S1 induced ROS generation and disturbed mitochondrial membrane potential. (**A**) Intracellular ROS levels were quantified by measuring DCF-derived fluorescence. HepG-2 cells were treated with WB-H-S1 at the concentration of 200 µg/mL or with same volume of methanol (control) for 24 h. Cellular DCF-derived fluorescence intensity was observed by a fluorescence microscope. (**B**) The quantification of the intensity of DCF-derived fluorescence after 24 h treatment with WB-H-S1 at the concentration of 200 µg/mL or with methanol (control). (**C**) HepG-2 cells were treated with WB-H-S1 at a concentration of 200 µg/mL or with methanol (control) for 24 h. The change in the JC-red/green ratio was measured by a fluorescence microplate. Data are mean ± SEM, *n* = 3, * *p* < 0.05.

**Figure 6 marinedrugs-20-00284-f006:**
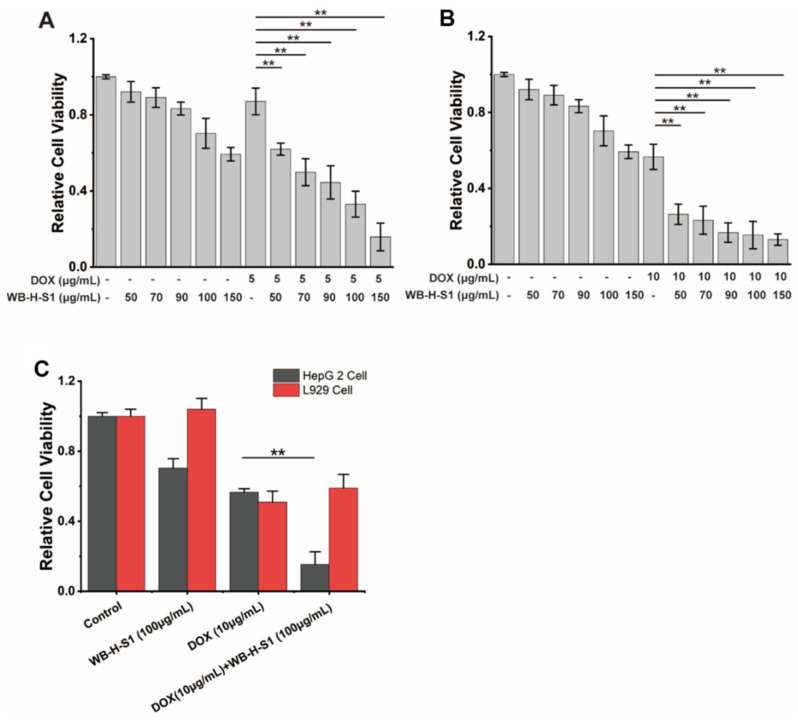
WB-H-S1 synergistically enhanced the anti-proliferative activity of doxorubicin. (**A**) HepG-2 cells were exposed to doxorubicin at the specified concentrations of 5 μg/mL for 48 h in the presence or absence of WB-H-S1 at a range of concentrations (50, 70, 90, 100, and 150 μg/mL), respectively. Relative cell viability was measured by MTT assay. (**B**) HepG-2 cells were exposed to doxorubicin at the specified concentrations of 10 μg/mL for 48 h in the presence or absence of WB-H-S1 at a range of concentrations (50, 70, 90, 100, and 150 μg/mL), respectively. The relative cell viability was measured by MTT assay. (**C**) HepG-2 cells and L929 cells were treated with doxorubicin at the concentrations of 10 μg/mL for 48 h in the presence or absence of WB-H-S1 at concentrations of 100 μg/mL. Relative cell viability was measured by MTT assay. Data are mean ± SEM, *n* = 3, ** *p* < 0.01.

**Figure 7 marinedrugs-20-00284-f007:**
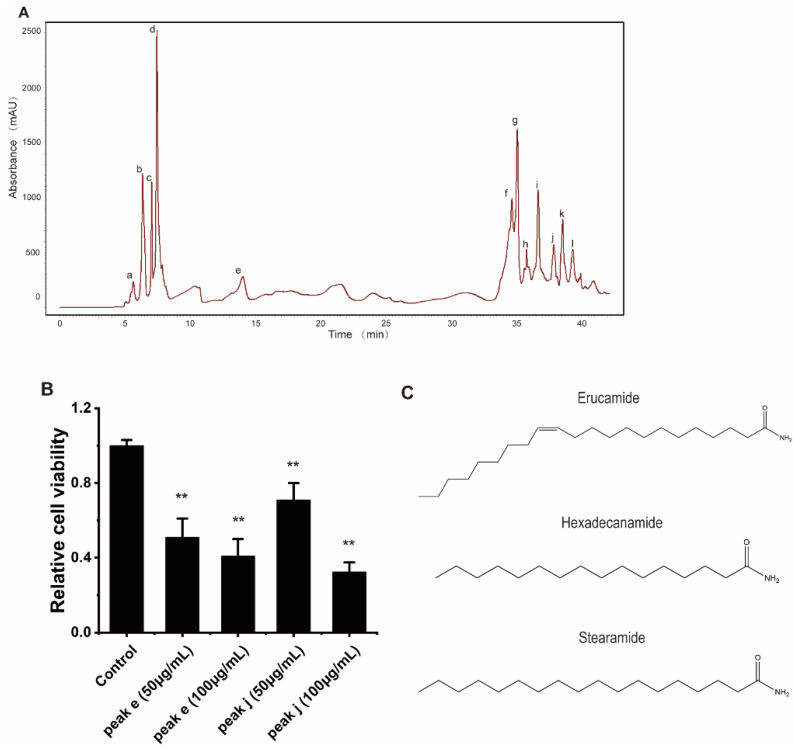
Isolation and identification of the active ingredients in WB-H-S1. (**A**) HPLC-DAD fingerprint of WB-H-S1 at 210 nm. HPLC led to the isolation of twelve ingredients (peak a–l) from WB-H-S1. (**B**) HepG-2 cells were treated with peak e or peak j at the concentration of 50 and 100 μg/mL for 48 h. Relative cell viability was measured by MTT assay. (**C**) Chemical structures of the identified compounds from peak j. Data are mean ± SEM, *n* = 3, ** *p* < 0.01.

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
