# Peer review of "Aqueous Extract of Sea Squirt (Halocynthia roretzi) with Potent Activity against Human Cancer Cells Acts Synergistically with Doxorubicin"

_marinedrugs, 2022, doi:10.3390/md20050284_

Round 1
Reviewer 1 Report
The manuscript entitled "Aqueous extract of Sea Squirt (Halocynthia roretzi) with Potent Activity against Human Cancer Cells Acts Synergistically with Doxorubicin" describes the discovery of three fatty acid amides as metabolites from sea quirt that act synergistically with doxorubicin as anticancer compounds.
Here are the major comments:
- It is odd to isolate long chain fatty acid amides from the aqueous extract.
- The figures were presented in a confusing manner; there is left and right figures not represented or not annotate well, e.g., alphabet. There are confusing figures in Figure 4, 5, and 6. Annotate the figures in letters.
- The best research approach for synergism effect is to test the IC50 of both compounds, rather than adding random test concentration and observe for significant decrease in cell viability. Retest the synergism assay by combining the the fatty acid amides with doxorubicin at their IC50 concentrations.
- Discuss the anticancer activity of fatty acid amides. What specific cancer cells do they target?
- The English language needs improvement. Kindly seek help or assistance on this matter.
Minor comment.
- Indicate the number of cells per well that was used in the following assay.
- MTT assay
- Apoptotic Assay
- ROS assay
- Mitochondrial Membrane potential
- It is not clear what database was used to determine the putative identities of the compounds based on MS/MS.
- Include in the supporting information the MS/MS spectra including the match spectra in the data base.
- How much fatty acid amides were obtained from 3 grams of the extract?
Author Response
Point-by-point response
The manuscript entitled "Aqueous extract of Sea Squirt (Halocynthia roretzi) with Potent Activity against Human Cancer Cells Acts Synergistically with Doxorubicin" describes the discovery of three fatty acid amides as metabolites from sea quirt that act synergistically with doxorubicin as anticancer compounds.
Here are the major comments:
- It is odd to isolate long chain fatty acid amides from the aqueous extract.
Response: Thanks for the question. We also feel odd when the identification results come out. However, our identification results were already confirmed by comparing with the reference standard substances. The possible reasons that we speculated are: 1. Solution of aqueous extracts and dialysis were not completely limpid. The solutions thus might contain long chain contain fatty amides, which then were absorbed on silica gel, and firstly eluded by methanol and ethyl acetate during silica gel column chromatography (WB-H-S1 was the first eluted fraction). 2. In the process of extraction, chemical reaction might occur between long chain fatty acid amides and other organic salt. So that they dissolved in water. 3. Small dose of fatty amides dissolved into the aqueous solution and were continuously enriched until they were detected.
- The figures were presented in a confusing manner; there is left and right figures not represented or not annotate well, e.g., alphabet. There are confusing figures in Figure 4, 5, and 6. Annotate the figures in letters.
Response: Sorry for the confusion. We have modified figure 4, 5, and 6, respectively, and the relevant panels have been labeled with letters. The caption of each figure has been re-edited carefully.
- The best research approach for synergism effect is to test the IC50 of both compounds, rather than adding random test concentration and observe for significant decrease in cell viability. Retest the synergism assay by combining the the fatty acid amides with doxorubicin at their IC50 concentrations.
Response: Thanks for the suggestion. We performed MTT experiments using gradient concentrations of the compounds from peak e and peak j, and calculated the IC50 value based on the standard curve. The results showed that the IC50 concentrations of mixture compounds from peak e and peak j, and doxorubicin to HepG 2 cells were 55 μg/mL, 77 μg/mL, and 15 μg/mL, respectively. Then we re-performed the synergism assay by combining the compounds from peak e and peak j (fatty acid amides) with doxorubicin, respectively, at their IC50 concentrations. The results have been added as Figure S2B in the revised version.
- Discuss the anticancer activity of fatty acid amides. What specific cancer cells do they target?
Response: It has been reported that polyunsaturated fatty acids (PUFA), such as DHA and EPA have the potential effects against diverse types of cancers. DHA has been demonstrated to be an anti-proliferative effect for human MDA-MB-231 breast cancer cell (Blanckaert, et al. 2010). EPA exhibited cytotoxic effect on colorectal cancer cell lines: Caco-2, HT-29, HCT116, LoVo, SW480, and SW620 (Jóźwiak, et al. 2020). Polyunsaturated fatty acid amides (PUFA amides), the active product of fatty acid hydrolysis, show similar anticancer activities. Some PUFA amides like N-docosahexaenoylethanolamine and N-eicosapentaenoylethanolamine exert immune-modulating effects and inhibition on breast cancer growth in vitro and in vivo(Giordano, et al. 2020). Fatty acid amides derived from oleic acid, palmitic acid, EPA, and DHA induced cell apoptosis in the SKOV-3 ovarian cancer cell line (El-Baz, et al. 2021). Fatty acid amide derivatives of the conventional anticancer drug doxorubicin show more lipophilic and exhibit more anti-proliferative activity for ovarian and colon cancer cell lines (Chhikara, et al. 2011).
Above discussion information has been added in the main text between line 340 and line 371.
- The English language needs improvement. Kindly seek help or assistance on this matter.
The English has been carefully examined by an invited language expert.
Minor comment.
- Indicate the number of cells per well that was used in the following assay.
- MTT assay
- Apoptotic Assay
- ROS assay
- Mitochondrial Membrane potential
Response: In MTT assay, approximately 1X104 cells per well were inoculated in 96-well plate. In apoptotic assay, approximately 1X106 cells per wells were inoculated with 6-well plate. 1X105 cells per well were inoculated with 24-well plate in ROS and mitochondrial membrane potential assays. This information has been added in Material and method part.
- It is not clear what database was used to determine the putative identities of the compounds based on MS/MS.
- Include in the supporting information the MS/MS spectra including the match spectra in the data base.
Response: We want to clarify that we didn’t use the database to determine the compounds. Instead, we identified the compounds from peak j by UHPLC-MS/MS through comparison of retention times and fragment ions with standard substances. Standard substances of hexadecanamide, stearamide, and erucamide (Aladdin, Shanghai) were used to determine the fragment ions and retention times of compounds in UHPLC-MS/MS. The precursor ion at m/z 256.3 and the product ion at m/z 88.0 were selected for the identification of hexadecanamide from extracts. The precursor ion at m/z 284.3 and the product ion at m/z 88.0 were selected for the identification of stearamide from aqueous extracts. The precursor ion at m/z 338.4 and the product ion at m/z 83.1 were selected for the identification of erucamide from aqueous extracts. The retention time of the compounds from aqueous extracts were consistent with that from the corresponding standard substances (Figure S3).
- How much fatty acid amides were obtained from 3 grams of the extract?
Response: 2 mg of fatty acid amides (peak j) were obtained from 3 g dialytic small molecular components of the aqueous extracts.
References
- Blanckaert V, et al. Docosahexaenoic acid intake decreases proliferation, increases apoptosis and decreases the invasive potential of the human breast carcinoma cell line MDA-MB-231. Int J Oncol. 2010.
- Jóźwiak M, et al. Anticancer activities of fatty acids and their heterocyclic derivatives. Eur J Pharmacol. 2020.
- Giordano C, et al. n-3 Polyunsaturated Fatty Acid Amides: New Avenues in the Prevention and Treatment of Breast Cancer. Int J Mol Sci. 2020.
- El-Baz HA,et al. Single Cell Oil (SCO)-Based Bioactive Compounds: I-Enzymatic Synthesis of Fatty Acid Amides Using SCOs as Acyl Group Donors and Their Biological Activities. Appl Biochem Biotechnol. 2021.
- Chhikara BS, et al. Fatty acyl amide derivatives of doxorubicin: synthesis and in vitro anticancer activities. Eur J Med Chem. 2011.
Reviewer 2 Report
The manuscript by Zhu et al, reports on the antitumor effects of the aqueous extract from the solitary ascidian Halocynthia roretzi, an edible animal. The manuscript is of undoubted interest to Mar. Drugs readers. However, in my opinion, it suffers from some inaccuracies in the text, especially for what concerns a clear presentation of the used methods and the rationale of the experiments.
Here below my comments
Major:
- Authors should explain better that they used 5 tumor cell lines and list them, and specify that L929 is not a tumor cell line in order to render the text of clear comprehension also to readers without familiarity with mammalian cell lines. In addition, they should explain why, after a single experiment with all the cell lines, they decided to focus with a single cell line (HepG2).
- The dialysis method is not well explained: which the used membrane? Which cutoff? This also to better understand what “small/macromolecular compounds” means.
- Para 4.6. The conditions of the flow cytometry are not adequately explained
- Para 4.7. Authors should describe which protease and phosphatase inhibitors were used and at which concentration
- Para 4.10. Authors should explain how was the solvent removed.
- line 138. fig. 3D: What does “cell density” means? How is it measured? It looks a measure cell proliferation rather than a measure of cell viability.
Minor:
line 45. Replace “provide” with “represent”
line 49. Ecteinascidia turbinate
line 66. “…roretzi, found to have antioxidant…”; “…roretzi, found to have antimicrobial…”
line 83. delete “there”
line 84. replace “detected” with “analyzed”
line 90. replace “part” with “fraction”. Insert “As” before “shown”
line 94. “…small molecule sample, at the concentration of 1 and 2 mg/ml, reduced…”
line 127. replace “on the contrary” with “conversely”
line 128. delete “the” before “HepG2”; replace “were” with “was”
line 147. The abbreviation for “Annexin-V FITC was already introduced and can be used. Therefore, replace “Annexin-V FITC” with “AV”
line 148. “2.90%, whereas the treatment …“
line 151 Fig. 3B at the end of the sentence?
line 154. replace “compare” with “compared”
line 157. “…induced apoptosis in HepG2 cells.”
line 159. replace “activated” with “activating”
line 169. “(C). Viability of HepG2 cells. Cells were…”
line 170. The abbreviation “PFTa” was never introduced before.
line 190. What concentration of methanol was used in controls?
line 230.delete “the” before “samples”
line 267. “apoptosis” (without capital initial)
line 268. “Sargassum” (with capital initial)
line 269. replace “induced” with “inducing”
line 270. replace “the” with “and”
line 272. replace “confirmed” with “confirming”
line 275. “…proteins, plays critical…”
line 277. replace “reduced” with “reducing”
line 291. replace “triggers” with “triggering”
line 292. replace “for other side” with “on another side”
line 293. replace “detected” with “measured2
line 294. Delete “the” before “WB-H-S1”
line 297. replace “shown” with “showed”
line 306. “…revealed that erucamide, and not hexadecanamide or stearamide,…”
line 309. insert “useful” before “against”
line 316. replace “about” with “related to the”
line 318. delete “are the first”
line 319. replace “investigated” with “demonstrated”
line 323. “…one of the reasons why extracts…”
line 344. “…tissue was decocted…”
line 349. delete “the” before “HeLa”
line 350. delete “the” before “HepG2”
line 351. delete “the” before “B16F1”
line 353. CO2: 2 subscript
line 357. delete “the” before “cells”
line 358. delete “the” before “medium”
line 363. delete “subject with”
line 372. “washed twice”
line 382. delete “the” before “membranes”
line 389. “permeable probe”. “which is converted”. Delete “into” (it appears twice”
line 392. delete “the” before “cells”
line 401. “at the concentration …”
line 403. delete “the” before “cells”; insert “were” before “incubated”; “at 37°C, in the incubator…”
line 404. “Cells were washed twice …”
Author Response
Point-by-point response
The manuscript by Zhu et al, reports on the antitumor effects of the aqueous extract from the solitary ascidian Halocynthia roretzi, an edible animal. The manuscript is of undoubted interest to Mar. Drugs readers. However, in my opinion, it suffers from some inaccuracies in the text, especially for what concerns a clear presentation of the used methods and the rationale of the experiments.
Here below my comments
Major:
- Authors should explain better that they used 5 tumor cell lines and list them, and specify that L929 is not a tumor cell line in order to render the text of clear comprehension also to readers without familiarity with mammalian cell lines. In addition, they should explain why, after a single experiment with all the cell lines, they decided to focus with a single cell line (HepG2).
Response: Five tumor cell lines that we used in the experiments are human hepatocellular carcinoma HepG 2, human melanoma cell BF16F1, human fibrosarcoma cell HT-1080, human thyroid anaplastic carcinoma cell BHT-101, and human cervical carcinoma cell HeLa. The reason that we used HepG 2 cells not the other cell lines to perform screening and functional assay is that the response of HepG 2 cells to the aqueous extracts was more pronounced compared with others.
- The dialysis method is not well explained: which the used membrane? Which cutoff? This also to better understand what “small/macromolecular compounds” means.
Response: Biotech Cellulose Ester (CE) Membrane (MWCO: 100 Da, Spectrumlabs, American) were used to do dialysis experiment. Small molecular compounds refer to external solution of dialysis tubing after dialysis overnight. Macromolecular compounds refer to internal solution of dialysis tubing after dialysis overnight. The dialysis method was added in the part of Materials and methods.
- Para 4.6. The conditions of the flow cytometry are not adequately explained
Response: The cells stained with PI/AV were analyzed using a flow cytometer (Accuri C6). A total of 30,000 cells were collected in flow cytometry. 488-nm laser were used on the flow cytometer for exciting fluorescence of AV/PI. Running of an untreated sample and adjustment of the voltage and gain for the FITC and PI detectors so that all cells can be detected in the bottom left quadrant. Running of a treated sample stained with FITC alone and adjustment of the voltage and gain for the FITC detector so that the dead cells appear in the bottom right quadrant. Running of a treated sample stained with PI alone and adjustment of the voltage and gain for the PI detector so that the dead cells appear in the top left quadrant. Adjustment of the compensation so that the live cells appear in the bottom left, the early apoptotic cells appear in the bottom right, late apoptotic appear in the top right quadrants, and the necrotic cells appear in the left right quadrants. The conditions of the flow cytometry were added in the part of Materials and methods.
- Para 4.7. Authors should describe which protease and phosphatase inhibitors were used and at which concentration
Response: Protease and phosphatase inhibitor cocktail (cat. no. P1046-1, Beyotime Institute of Biotechnology, Shanghai) were used in protein extraction. Protease and phosphatase inhibitor was added to the lysate in a ratio of 1:50. This information was added in the part of Materials and methods.
- Para 4.10. Authors should explain how was the solvent removed.
Response: Solvent of fractions divided by semipreparative-HPLC were evaporated to dryness under 55℃. The methods of solvent removing were added in 4.11 Isolation and identification of active ingredients.
- line 138. fig. 3D: What does “cell density” means? How is it measured? It looks a measure cell proliferation rather than a measure of cell viability.
Response: Cell density means the density of living cells in each well after treatment with the extracts or corresponding solvent. MTT assay was used to measure the density of living cells in each well.
What concentration of methanol was used in controls?
Response: The added volume of methanol was equal to the highest dose of the experimental group.
Minor:
line 45. Replace “provide” with “represent”
line 49. Ecteinascidia turbinate
line 66. “…roretzi, found to have antioxidant…”; “…roretzi, found to have antimicrobial…”
line 83. delete “there”
line 84. replace “detected” with “analyzed”
line 90. replace “part” with “fraction”. Insert “As” before “shown”
line 94. “…small molecule sample, at the concentration of 1 and 2 mg/ml, reduced…”
line 127. replace “on the contrary” with “conversely”
line 128. delete “the” before “HepG2”; replace “were” with “was”
line 147. The abbreviation for “Annexin-V FITC was already introduced and can be used. Therefore, replace “Annexin-V FITC” with “AV”
line 148. “2.90%, whereas the treatment …“
line 151 Fig. 3B at the end of the sentence?
line 154. replace “compare” with “compared”
line 157. “…induced apoptosis in HepG2 cells.”
line 159. replace “activated” with “activating”
line 169. “(C). Viability of HepG2 cells. Cells were…”
line 170. The abbreviation “PFTa” was never introduced before.
line 190. What concentration of methanol was used in controls?
line 230.delete “the” before “samples”
line 267. “apoptosis” (without capital initial)
line 268. “Sargassum” (with capital initial)
line 269. replace “induced” with “inducing”
line 270. replace “the” with “and”
line 272. replace “confirmed” with “confirming”
line 275. “…proteins, plays critical…”
line 277. replace “reduced” with “reducing”
line 291. replace “triggers” with “triggering”
line 292. replace “for other side” with “on another side”
line 293. replace “detected” with “measured2
line 294. Delete “the” before “WB-H-S1”
line 297. replace “shown” with “showed”
line 306. “…revealed that erucamide, and not hexadecanamide or stearamide,…”
line 309. insert “useful” before “against”
line 316. replace “about” with “related to the”
line 318. delete “are the first”
line 319. replace “investigated” with “demonstrated”
line 323. “…one of the reasons why extracts…”
line 344. “…tissue was decocted…”
line 349. delete “the” before “HeLa”
line 350. delete “the” before “HepG2”
line 351. delete “the” before “B16F1”
line 353. CO2: 2 subscript
line 357. delete “the” before “cells”
line 358. delete “the” before “medium”
line 363. delete “subject with”
line 372. “washed twice”
line 382. delete “the” before “membranes”
line 389. “permeable probe”. “which is converted”. Delete “into” (it appears twice”
line 392. delete “the” before “cells”
line 401. “at the concentration …”
line 403. delete “the” before “cells”; insert “were” before “incubated”; “at 37°C, in the incubator…”
line 404. “Cells were washed twice …”
Thank you for the careful reading of the manuscript. All the grammar errors and typos have been corrected in the revised version.
Round 2
Reviewer 1 Report
The manuscript has improved and the authors addressed the concerns and suggestions of the reviewers.
Author Response
Thanks for the reviewer
Reviewer 2 Report
Although improved, the manuscript still requires some revisions. Here below my comments.
line 35. "...leads ultimately to the damage..."
line 39. "...from marine organisms with a wide..."
line 40. Delete the comma after "structures"
line 42. "...marine organisms. The most abundant sources of active products are fungi..."
line 52. "...from ocean animals and was approved..."
line 54. "..albicans, is currently..."
line 58. "Therefore investigation on ascidians is important for the discovery..."
line 62. ...the esatern and southern coasts of what region? Asia? Korea? Please, specify.
line 66. "...such as peptides and hemagglutinin with antoimicrobial activity [15, 16]. Hence..."
line 72. "broad spectrum"
line 7a. ""...we have characterized and fractionated..."
line 76. "ascidians"
line 123. "...showed a significant..."
line 161 and other. Please, use the lowercase "p" for "p53"
line 165. "...the mortality indiced by WB..."
line 173. "Western"
line 177. The abbreviation "ROS" was never introduced before. Even if widely used, its meaning (reactive oxygen species) should be introduced.
line 178. Delete "in cells"
line 179. "play"
line 180. Replace "different" with "various"
line 203. "DOX is a topoisomerase..."
line 206. "...clinical, the use of DOX has been limited by the ...."
line 213. Replace "highly" (without a statistical meaning) with "significantly"
line 214. "...enhanced. When HepG2 cells and L929 cells were treated with WB-H-S1 at the concentration of 100 µg/mL, for 48h, in the presence or absence of DOX, a significamtly higher cytotoxicity towards HepG2 cells than L929 cells was obserces (figure 6C)"
line 218. Replace "acted" with "acts2
line 219. "reduces2
line 235. "...systematic fractionation ..."
line 283. "The balance between..."
line 286. "found" instead of "find"
line 293. "...cells. Recently, induced oxidative stress causing severe damages in..."
line 294. "processes, was proposed as a parameter to be considered in antitumor drug screening..."
line 300. "...activation ans cellular ROS are modulated by p53 in many ways..."
line 303. "...WB-H-S1. As expected, WB-H-S1 increased ...level and cells mortality was reduced in the presence of p53 inhibitors.
line 305. "...thsi is likely the main reason of the observed low toxicity in normal cells."
Line 321-322. The sentence does not explain the different toxicity of WB-H-S1 towards tumot and normal cells as in both cases hexadecanamide and stearamide are present.
Author Response
Point-by-point response
Although improved, the manuscript still requires some revisions. Here below my comments.
line 35. "...leads ultimately to the damage..."
line 39. "...from marine organisms with a wide..."b
line 40. Delete the comma after "structures"
line 42. "...marine organisms. The most abundant sources of active products are fungi..."
line 52. "...from ocean animals and was approved..."
line 54. "..albicans, is currently..."
line 58. "Therefore investigation on ascidians is important for the discovery..."
line 62. ...the esatern and southern coasts of what region? Asia? Korea? Please, specify.
line 66. "...such as peptides and hemagglutinin with antoimicrobial activity [15, 16]. Hence..."
line 72. "broad spectrum"
line 7a. ""...we have characterized and fractionated..."
line 76. "ascidians"
line 123. "...showed a significant..."
line 161 and other. Please, use the lowercase "p" for "p53"
line 165. "...the mortality indiced by WB..."
line 173. "Western"
line 177. The abbreviation "ROS" was never introduced before. Even if widely used, its meaning (reactive oxygen species) should be introduced.
line 178. Delete "in cells"
line 179. "play"
line 180. Replace "different" with "various"
line 203. "DOX is a topoisomerase..."
line 206. "...clinical, the use of DOX has been limited by the ...."
line 213. Replace "highly" (without a statistical meaning) with "significantly"
line 214. "...enhanced. When HepG2 cells and L929 cells were treated with WB-H-S1 at the concentration of 100 µg/mL, for 48h, in the presence or absence of DOX, a significamtly higher cytotoxicity towards HepG2 cells than L929 cells was obserces (figure 6C)"
line 218. Replace "acted" with "acts2
line 219. "reduces2
line 235. "...systematic fractionation ..."
line 283. "The balance between..."
line 286. "found" instead of "find"
line 293. "...cells. Recently, induced oxidative stress causing severe damages in..."
line 294. "processes, was proposed as a parameter to be considered in antitumor drug screening..."
line 300. "...activation ans cellular ROS are modulated by p53 in many ways..."
line 303. "...WB-H-S1. As expected, WB-H-S1 increased ...level and cells mortality was reduced in the presence of p53 inhibitors.
line 305. "...thsi is likely the main reason of the observed low toxicity in normal cells."
Response: Thank you very much for the careful reading of the manuscript. All the above grammar errors and typos have been corrected in the revised version.
Line 321-322. The sentence does not explain the different toxicity of WB-H-S1 towards tumot and normal cells as in both cases hexadecanemide and stearamide are present.
Response: Based on our results, we speculated that the low toxicity of WB-H-S1 to normal cells might be related to the ability of WB-H-S1 to induce the increase of the intracellular ROS levels. High ROS level in cancer cells promotes cancer development by participating in signaling pathways (Yang, et al. 2016). Therefore, tumor cells are more sensitive to the change in ROS levels compared with the normal cells. In addition, molecules except for fatty acid amides in WB-H-S1 may possess protective effects on normal cell. As the sentence “The antitumor activity and low toxicity to L929 cells from this mixture sample may be caused by the protection of hexadecanamide or stearamide and the toxicity of erucamide” could not explain the different toxicity of WB-H-S1 towards tumor and normal cells, we have removed this sentence in the revised version.
Reference
- Yang Y, Karakhanova S, Hartwig W, D' Haese JG, Philippov PP, Werner J, Bazhin AV. Mitochondria and Mitochondrial ROS in Cancer: Novel Targets for Anticancer Therapy. J Cell Physiol. 2016 Dec;231(12):2570-81.